# Sparsing Law: Towards Large Language Models with Greater Activation Sparsity

**Yuqi Luo** [* 1]  **Chenyang Song** [* 1]  **Xu Han** [1]  **Yingfa Chen** [1]  **Chaojun Xiao** [1]  **Xiaojun Meng** [2]  **Liqun Deng** [2]
**Jiansheng Wei** [2]  **Zhiyuan Liu** [1]  **Maosong Sun** [1]

{luo-yq23,scy22}@mails.tsinghua.edu.cn, {han-xu,liuzy}@tsinghua.edu.cn

## Abstract

Activation sparsity denotes the existence of substantial weakly-contributed neurons within feed-forward networks of large language models (LLMs), providing wide potential benefits such as computation acceleration. However, existing works lack thorough quantitative studies on this useful property, in terms of both its measurement and influential factors. In this paper, we address three underexplored research questions: (1) How can activation sparsity be measured more accurately? (2) How is activation sparsity affected by the model architecture and training process? (3) How can we build a more sparsely activated and efficient LLM? Specifically, we develop a generalizable and performance-friendly metric, named CETT-PPL-1%, to measure activation sparsity. Based on CETT-PPL-1%, we quantitatively study the influence of various factors and observe several important phenomena, such as the convergent power-law relationship between sparsity and training data amount, the higher competence of ReLU activation than mainstream SiLU activation, the potential sparsity merit of a small width-depth ratio, and the scale insensitivity of activation sparsity. Finally, we provide implications for building sparse and effective LLMs, and demonstrate the reliability of our findings by training a 2.4B model with a sparsity ratio of 93.52%, showing 4.1× speedup compared with its dense version. The codes and checkpoints are available at https://github.com/thunlp/SparsingLaw.

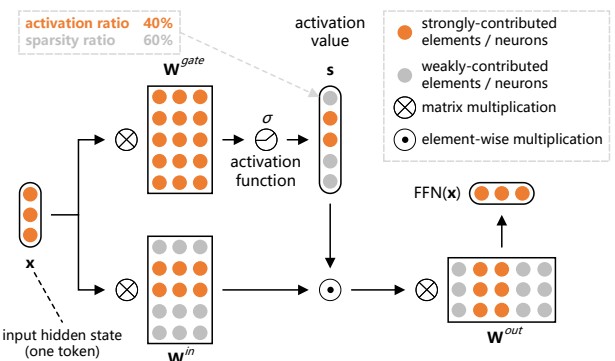

Figure 1: A typical case of activation sparsity (with a sparsity ratio of 60%) in a gated feed-forward network of LLMs.

## 1. Introduction

Activation sparsity refers to the phenomenon that considerable elements within the output of activation layers (shown in Figure 1) are zero or low values, and thus the corresponding neurons[1] contribute weakly to the final model output given a specific input. As a prevalent property of many language and vision modeling architectures (Li et al., 2022), activation sparsity has wide practical values, such as inference acceleration (Liu et al., 2023; Song et al., 2023), training acceleration (Zhang et al., 2024b), and LLM interpretation (Sajjad et al., 2022). For instance, Xue et al. (2024) achieves up to 27.8× inference acceleration on smartphones by utilizing activation sparsity, while Zhang et al. (2023) reveals and interprets the emergent modularity in LLMs by analyzing the specialization of sparsely-activated neurons.

Despite the extensive benefits of activation sparsity, there are few relevant comprehensive and quantitative studies. To fill this gap, we focus on three research questions:

- (Q1) How can activation sparsity be measured more accurately?

- (Q2) How is activation sparsity affected by the model architecture and training process?

---

[*]Equal contribution  [1]Dept. of Comp. Sci. & Tech., Institute for AI, Tsinghua University, Beijing, China [2]Huawei Noah's Ark Lab, China. Correspondence to: Xu Han <han-xu@tsinghua.edu.cn>, Zhiyuan Liu <liuzy@tsinghua.edu.cn>.

*Proceedings of the 42nd International Conference on Machine Learning*, Vancouver, Canada. PMLR 267, 2025. Copyright 2025 by the author(s).

---

[1]A neuron denotes a certain row or column within the parameter matrices of feed-forward networks (FFN).

- (Q3) How can we build a more sparsely activated and efficient LLM?

**Metric for Activation Sparsity**  A "metric" for activation is for measuring the sparsity ratio (i.e., the ratio of weakly-contributed neurons) of an LLM given specific inputs. A good metric should be generalizable (i.e., applicable to different activation functions), and its impact on the model performance (after the neurons recognized as weakly-contributed ones are skipped during calculation) should be negligible. To achieve this, we need an accurate algorithm to identify weakly-contributed neurons in each layer. The more accurate this algorithm is, the less performance degradation a model will suffer under a specific sparsity ratio.

We experimentally demonstrate that CETT (Zhang et al., 2024a), a generalizable method adaptively setting layer-wise thresholds for activation values to identify weakly-contributed neurons (Section 2.2), achieves the best trade-off between performance and sparsity ratio among existing methods. To further ensure negligible performance degradation, we develop **CETT-PPL-**$1\%$ as a better sparsity metric. Specifically, this metric binary searches the CETT hyper-parameter (controlling the output error of FFNs in each layer after weakly-contributed neurons are skipped) under a PPL increase tolerance of 1% that causes nearly no harm to performance on downstream tasks. In the 0.8B ReLU-activated model (Figure 2), it reduces the activation ratio by 61% relatively compared with the common ReLU-oriented metric using a zero threshold.

**Influential Factors of Activation Sparsity**  Using CETT-PPL-1%, we systematically study the correlation between activation sparsity and four influential factors, including **the amount of pre-training data, the activation function, the width-depth ratio (i.e., the ratio of the hidden dimension to the layer number), and the parameter scale**. Through comprehensive experiments on models scaled from 0.1B to 1.2B, we obtain the following observations:

1. There is an increasing power-law (mainstream SiLU-activated LLMs) or decreasing logspace power-law (ReLU-activated LLMs) relationship between the activation ratio ($1 - \text{sparsity ratio}$) and the amount of pre-training data. Both laws are convergent with a certain limit sparsity ratio as the amount of data approaches infinity. ReLU-activated LLMs can be more sparsely activated with more data. (Figure 4)

2. Given the same parameter scale, the sparsity ratio of ReLU-activated LLMs always surpasses that of SiLU-activated LLMs, with comparable performance on downstream tasks. (Figure 4 and Table 1)

3. Given the same parameter scale, the activation ratio linearly increases with the width-depth ratio under a bottleneck (i.e., deeper models are sparser), above which activation fluctuates around a fixed level. (Figure 5)

4. Given similar width-depth ratios, the limit of activation sparsity is weakly correlated to the scale of LLMs, while the convergence speed to the limit is much faster in smaller models. (Figure 7) We try to explain these phenomena in Section 5.3, indicating a similar neuron specialization pattern across models of distinct scales.

**Approach to More Sparsely-activated LLMs**  Based on the above findings, we answer the third question: To train an LLM with greater activation sparsity from scratch, it is better to adopt the ReLU activation, use a small width-depth ratio on the premise of stable training, and feed more training data so that the decreasing logspace power-law can help reduce the activation ratio.

To validate the generalizability of our findings, we train a larger 2.4B model. With ReLU activation, near 800B training tokens, and a width-depth ratio close to the 0.1B-1.2B experimental models (which is small enough and ensures training stability), the model undergoes a similar logspace power-law trend between activation and data, achieving a high limit sparsity ratio of 93.52% and $4.1\times$ speedup ratio to the dense model. This value is also close to the limit sparsity ratios of 0.1B-1.2B models, consistent with the previously found scale insensitivity of the limit sparsity.

To sum up, our work provides a better practice of measuring activation sparsity and then comprehensively studies its influential factors and scaling properties. The empirical laws found above can provide instructional values for designing and pre-training an LLM with greater activation sparsity, which helps produce more efficient LLMs.

## 2. Preliminaries and Related Works

### 2.1. Preliminaries of Activation Sparsity

Activation sparsity is a prevalent property existing in neural networks with activation layers, indicating the existence of considerable neurons, that correspond to zero or low activation values, having a limited impact on final network outputs given specific inputs. Mainstream LLMs present remarkable activation sparsity (Li et al., 2022; Zhang et al., 2022; Song et al., 2025; 2024).

Activation sparsity can help improve LLMs in many aspects, such as efficiency and interpretability. Recent works manage to exploit activation sparsity for inference acceleration, mainly by saving the computation related to weakly-contributed parameters (Liu et al., 2023; Song et al., 2023; Xue et al., 2024). Zhang et al. (2024b) utilize the existence of activation sparsity throughout the majority of the LLM pre-training process to accelerate pre-training. Besides acceleration, activation sparsity also helps improve interpretability, which is important for reliable and well-performing LLMs. Explanation of the sparse neuron activation patterns has

long been a mainstream paradigm of interpreting LLM behaviors (Sajjad et al., 2022; Bills et al., 2023; Gao et al., 2024; Lieberum et al., 2024).

Notably, mixture-of-experts (MoE) is a popular paradigm of activation sparsity, enforcing each token to activate a limited number of experts. As such constraints can sacrifice flexibility and performance (see Appendix B), we focus on activation sparsity in vanilla Transformers in this work.

Besides, activation sparsity also has fundamental differences from weight pruning, which realizes acceleration by removing certain parts of LLM parameters regardless of inputs. By contrast, activation sparsity adaptively determines important neurons for different inputs with no parameter permanently removed, causing considerably less performance degradation (see Appendix C).

## 2.2. Metrics of Activation Sparsity

Building a satisfactory metric for measuring sparsity is a nontrivial work. For the convenience of demonstrations, we formally introduce the following notations for the computation process of FFNs (also see Figure 1). With a hidden dimension $d_h$ and an intermediate dimension $d_f$, a gated FFN (the FFN paradigm in mainstream LLMs (Dauphin et al., 2017; Shazeer, 2020)) works as follows:

$$\mathbf{s} = \sigma(\mathbf{W}^{gate}\mathbf{x}), \quad \text{FFN}(\mathbf{x}) = \mathbf{W}^{out}[\mathbf{s} \odot (\mathbf{W}^{in}\mathbf{x})], \quad (1)$$

where $\mathbf{x} \in \mathbb{R}^{d_h}$, $\mathbf{s} \in \mathbb{R}^{d_f}$, $\sigma$, and $\odot$ denote the input hidden states, the activation values, the activation function, and the element-wise multiplication, respectively. $\mathbf{W}^{gate}, \mathbf{W}^{in} \in \mathbb{R}^{d_f \times d_h}$ and $\mathbf{W}^{out} \in \mathbb{R}^{d_h \times d_f}$ are learnable parameters. Next, we decompose the parameters of FFN along the dimension of $d_f$ into $d_f$ neurons. The output of the $i$-th neuron $n_i$ is calculated by

$$s_i = \sigma(\mathbf{W}^{gate}_{i,:}\mathbf{x}), \quad n_i = \mathbf{W}^{out}_{:,i}[s_i \odot (\mathbf{W}^{in}_{i,:}\mathbf{x})],$$
$$\text{FFN}(\mathbf{x}) = \sum_{i=1}^{d_f} n_i, \quad (2)$$

where $\mathbf{W}^{gate}_{i,:}, \mathbf{W}^{in}_{i,:}, \mathbf{W}^{out}_{:,i}$ are the $i$-th row of $\mathbf{W}^{in}$, the $i$-th row of $\mathbf{W}^{gate}$, and the $i$-th column of $\mathbf{W}^{out}$, respectively. The FFN outputs equals the sum of all neuron outputs.

Activation sparsity is measured by the ratio of weakly-contributed neurons, namely $|\mathcal{D}|/d_f$, where $\mathcal{D}$ is the index set of weakly-contributed neurons. Different metrics mainly differ in determining whether a specific neuron contributes weakly. A straightforward metric, naturally adopted in ReLU-activated models, regards neurons with zero activation values as weakly-contributed, namely $\mathcal{D} = \{i|s_i = 0\}$. To find non-zero weakly contributed activations, Kurtz et al. (2020) and Mirzadeh et al. (2023) introduce a positive threshold or bias, i.e., $\mathcal{D} = \{i|s_i < \epsilon\}$, $\epsilon > 0$.

The major drawback of this straightforward definition is the lack of generalizability. Concretely, it is unsuitable for activation functions with considerable non-negligible negative outputs, such as SiLU (Elfwing et al., 2018). In these cases, the straightforward metric can lose considerable negative neuron outputs and harm performance. A quick fix is to use the absolute value, $\mathcal{D} = \{i||s_i| < \epsilon\}$, but a global threshold across layers is hard to determine. Zhang et al. (2024a) adaptively searches the layer-wise thresholds by introducing the cumulative errors of tail truncation (CETT). Defined as the $L_2$ norm relative error caused by skipping weakly-contributed neurons, CETT is computed as:

$$\text{CETT} = \frac{\|\sum_{i \in \mathcal{D}} n_i\|_2}{\|\text{FFN}(\mathbf{x})\|_2}, \quad \mathcal{D} = \{i|\|n_i\|_2 < \epsilon\}, \quad (3)$$

where $\|\cdot\|_2$ is the $L_2$ norm operator. As CETT increases monotonically with $\epsilon$, for each layer, we can use binary search to find the threshold $\epsilon$ to meet the CETT value.

In addition to the above sparsity metrics, we also note some special practices. For instance, CATS (Lee et al., 2024) finds layer-wise adaptive thresholds but targets the same expected sparsity ratio at each layer. This is equivalent to the Top-$k$ setting in Section 4.1. TEAL (Liu et al., 2024a) addresses a more general paradigm named input sparsity. This is largely different from output sparsity, which is the paradigm handled in this work and mainly induced by activation functions. We leave the quantitative studies on input sparsity for future work.

In this work, we experimentally demonstrate that CETT, when tuned to just make the validation PPL increase by 1%, is a metric with great generalizability and the least performance degradation, even at a high sparsity ratio.

## 2.3. Influential Factors of Activation Sparsity

Despite the importance of activation sparsity, few works conduct comprehensive and quantitative studies on how it is affected by the model architecture and training process. Speculating that activation sparsity comes from the training dynamic in the optimization process, Li et al. (2022) finds an increasing trend of sparsity with larger scale, depth, and width in T5 series (Raffel et al., 2020). Zhang et al. (2024a) dives into this problem from the aspect of activation functions. Song et al. (2025) discovers that LLMs tend to be sparser on more formatted datasets such as codes and multiple choices. Other works have discussed the scaling properties of parameter-sparse models (Frantar et al., 2023), fine-grained MoE models (Krajewski et al., 2024), and sparse autoencoders (Gao et al., 2024). To the best of our knowledge, we present the first comprehensive quantitative study on the accurate measurement, influential factors, and training practice of activation sparsity for LLMs.

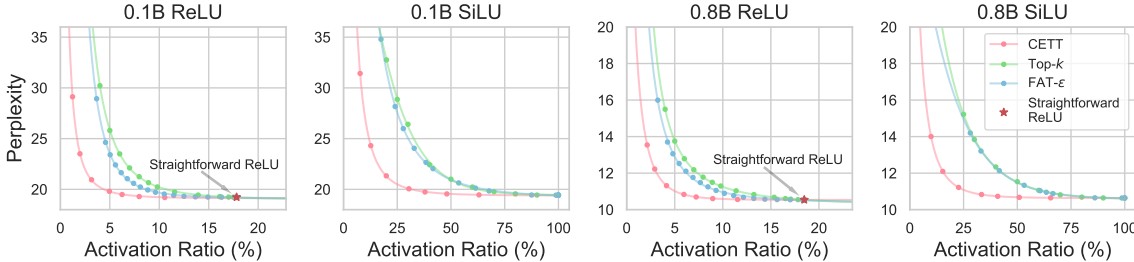

Figure 2: The PPL-activation ratio (i.e., $1 -$ sparsity ratio) Pareto curve using different methods to recognize weakly-contributed neurons. "Straightforward ReLU" is only applicable to ReLU-activated models using the zero threshold.

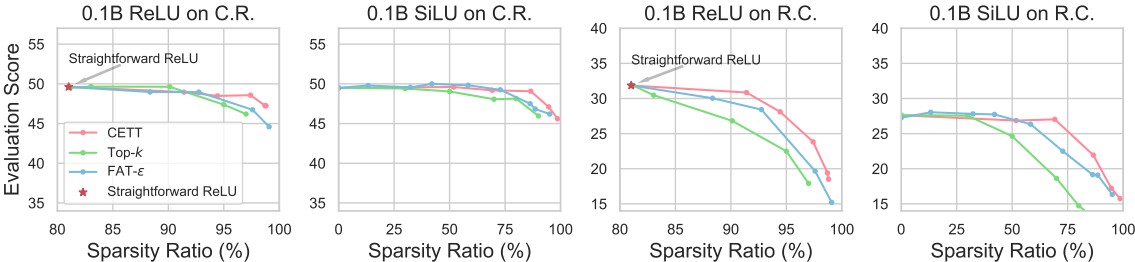

Figure 3: The Pareto curves between downstream task performance and activation sparsity using different methods to recognize weakly-contributed neurons. C.R. refers to *commonsense reasoning* and R.C. refers to *reading comprehension*.

## 3. Quantitative Study Settings

To address the three issues in the Introduction section, we conduct extensive experiments, training, evaluating, and analyzing the models ranging from 0.1B to 1.2B. We introduce our experimental settings in detail as follows.

**Model settings** We adopt the architecture of $\mu P$ Transformer (Hu et al., 2024), which combines the mainstream LLaMA (Touvron et al., 2023) with gated FFN and $\mu P$ parametrization (Yang et al., 2022) for training stability.

**Training settings** We mainly explore the activation sparsity of foundation models, which only undergo the pre-training stage. However, before we evaluate models on task-specific benchmarks, we follow recent LLMs (Dubey et al., 2024; Hu et al., 2024) to conduct a decay stage, where instruction-tuning data is added. Thereby, we can obtain more reasonable results on benchmarks. We also follow the optimal batch sizes and learning rates in recent LLMs.

**Evaluation settings** We introduce a tiny validation dataset and two groups of benchmarks for evaluation, including commonsense reasoning (C.R.) and reading comprehension (R.C.). For the measurement of sparsity, to eliminate the impact of stochastic factors (especially the sparsity fluctuations during the early stage), we employ a sparsity stabilizing strategy (see Appendix E).

If there are no special statements, the training loss, validation loss, and perplexity (on validation data) are all calcu-

lated on models that only complete pre-training, and the task-specific performance is evaluated on checkpoints after the decay stage. See Appendix H, I, and K for more details about datasets and training settings.

## 4. How can activation sparsity be measured more accurately?

### 4.1. Recognition of Weakly-Contributed Neurons

As stated above, the crucial part of a metric for activation sparsity is the accurate recognition of weakly-contributed neurons. In this section, we compare the following four methods to determine weakly-contributed neurons in FFNs (Refer to Eq. (2) for the definition of neurons):

(1) **Straightforward ReLU** is the most simple but commonly used setting, which uses a zero threshold and is only applicable to ReLU: $\mathcal{D} = \{i|s_i = 0\}$.

(2) **Top-$k$**, widely adopted in the MoE architectures (Fedus et al., 2022), enforces each layer to consistently maintain $k$ activated neurons, whose absolute activation values rank in the top-k ones among all the neurons of that layer. Obviously, we have $|\mathcal{D}| = d_f - k$, and the Top-$k$ sparsity method holds a constant sparsity ratio across all layers.

(3) **FAT-$\epsilon$** (Kurtz et al., 2020) (FAT denotes forced activation threshold) similarly introduces a global hyper-parameter $\epsilon$ as the threshold shared by all layers, namely $\mathcal{D} = \{i||s_i| < \epsilon\}$. Note that this is slightly different from the original

Table 1: The average evaluation scores (%) on two task groups. The second column represents settings with different PPL increase tolerances $p\%$, with "Dense" indicating the most accurate case where $p = 0$. The marker "Δ" means the score difference to the corresponding dense setting.

| | | 0.1B | | 0.2B | | 0.4B | | 0.8B | | 1.2B | | Avg. |
|---|---|---|---|---|---|---|---|---|---|---|---|---|
| | | ReLU | SiLU | ReLU | SiLU | ReLU | SiLU | ReLU | SiLU | ReLU | SiLU | |
| C.R. | Dense | 49.6 | 49.5 | 52.0 | 52.2 | 54.7 | 55.8 | 56.8 | 57.6 | 60.0 | 59.6 | 54.78 |
| | Δ **CETT-PPL-1%** | −0.5 | +0.4 | −0.3 | +0.2 | −0.1 | 0 | −0.9 | 0 | −0.4 | 0 | **−0.16** |
| | Δ CETT-PPL-5% | −0.4 | −0.5 | −0.3 | −0.2 | −0.4 | −0.7 | −0.5 | −0.5 | −0.7 | −0.8 | −0.50 |
| | Δ CETT-PPL-10% | −0.2 | −0.8 | −0.4 | −0.3 | +0.2 | −0.6 | −0.2 | −1.2 | −0.7 | −0.3 | −0.45 |
| R.C. | Dense | 28.2 | 27.7 | 40.7 | 40.2 | 44.0 | 41.8 | 44.8 | 43.3 | 53.2 | 54.8 | 41.87 |
| | Δ **CETT-PPL-1%** | +0.2 | +0.3 | −1.0 | −0.6 | −1.1 | −0.9 | −1.6 | +1.0 | +0.1 | +0.6 | **−0.30** |
| | Δ CETT-PPL-5% | −1.3 | −1.2 | −2.1 | −3.4 | −3.2 | −3.6 | −2.6 | −2.6 | +0.1 | −2.2 | −2.21 |
| | Δ CETT-PPL-10% | −2.0 | −2.9 | −2.1 | −5.8 | −4.1 | −6.5 | −4.5 | −4.5 | −0.3 | −3.7 | −3.64 |

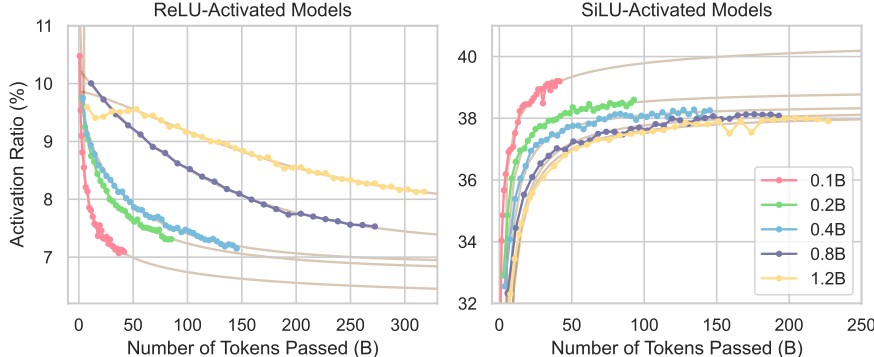

Figure 4: The trend of activation ratios of models with different parameter numbers and activation functions during the pre-training stage. The fitted curves are plotted in brown.

FATReLU by Kurtz et al. (2020) as we compute the absolute values of activation values to accommodate SiLU.

(4) **CETT** (Zhang et al., 2024a), as introduced in Section 2.2, requires each layer to share the same $L_2$ norm relative error after weakly-contributed neurons are skipped in calculation, while the layer-wise activation thresholds are adaptively searched accordingly.

As demonstrated by Figure 2 and Figure 3, **CETT obtains the best trade-off between sparsity and performance**. Under the same target sparsity ratio, CETT can always achieve the lowest PPL, and a comparable or higher evaluation score on downstream tasks. These indicate the highest accuracy of CETT in recognizing weakly-contributed neurons.

### 4.2. Introduction of PPL Increase Tolerance

Despite the advantages of CETT, it remains a problem how to choose the best hyper-parameter, namely the shared $L_2$ norm relative error. Therefore, we introduce CETT-PPL-$p\%$, which denotes the sparsity ratio measured by CETT when the PPL on validation data rises by $p\%$ compared with the dense setting (i.e., with all neurons activated). Given a PPL increase tolerance of $p\%$, we can conduct a binary search to find the hyper-parameter that just meets the tolerance. The search algorithm is described in Appendix F.

To find a proper $p\%$ tolerance, we inspect its impact on downstream task performance. As shown in Table 1, with PPL increasing (intrinsically promoting greater sparsity), the reading comprehension performance is considerably impaired, corresponding to the trade-off between sparsity and performance. Notably, in both task groups, the average performance of "CETT-PPL-1%" is comparable to that of the theoretically most accurate "Dense" setting. Therefore, **we assume CETT-PPL-1% sparsity as a reliable performance-friendly metric and employ it to compute sparsity in the following discussions**.

## 5. How is activation sparsity affected by the model architecture and training process?

### 5.1. Amount of Pre-Training Data and Choice of Activation Functions

To obtain the scaling relationship between the activation sparsity and the amount of pre-training data, we pre-train models with different parameter numbers and two activation functions (i.e., ReLU and SiLU), respectively, and then

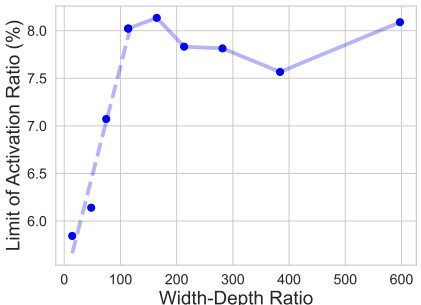

Figure 5: Limits of activation ratios on 0.1B ReLU models.

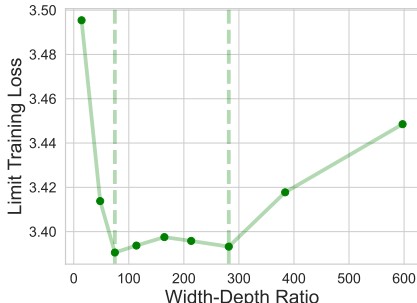

Figure 6: Limit training loss on 0.1B ReLU models.

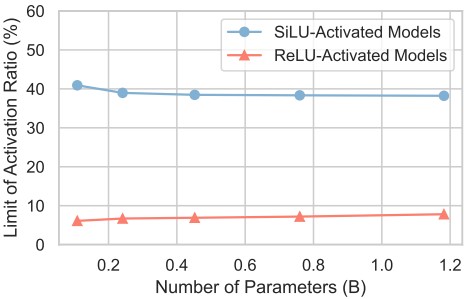

Figure 7: The limits of activation ratios on models with different scales and activation functions.

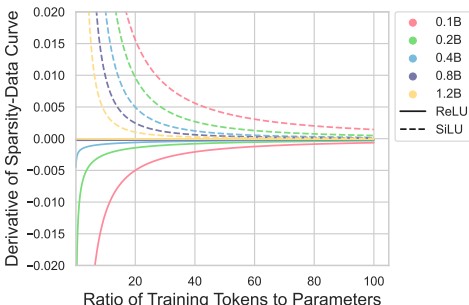

Figure 8: The derivative trends of the sparsity-data curve with the increase of data-scale ratio.

evaluate the sparsity level of their checkpoints using CETT-PPL-1%. After careful attempts, we find that the curve of activation ratios to the amount of pre-training data is easier to fit than that of sparsity ratios. Therefore, we will frequently use activation ratios instead of sparsity ratios in the following sections, whose trend is plotted in Figure 4.

For ReLU models, we observe a logspace power-law relationship between the activation ratio $A_{ReLU}(D)$ and the amount of pre-training data $D$, expressed as follows:

$$A_{ReLU}(D) = \exp(-cD^\alpha + b) + A_0, \qquad (4)$$

where $A_0 > 0$ is the limit of activation ratio with infinite training data and $c, \alpha > 0$. This is a convergent decreasing function, indicating that **more training data can potentially make ReLU models more sparsely-activated**.

By contrast, the activation ratio $A_{SiLU}(D)$ of SiLU models exhibit a vanilla power-law relationship:

$$A_{SiLU}(D) = -\frac{c}{D^\alpha} + A_0, \qquad (5)$$

where similarly, $A_0 > 0$ is the limit of activation ratio and $c, \alpha > 0$. Note that this is a convergent increasing function, and thus **more training data will impair the activation sparsity of SiLU models**. See Appendix G for the algorithm of curve fitting and the results (i.e., coefficients).

As for the selection of activation functions, by comparing the sparsity dynamics, we can conclude that the activation

sparsity achieved by ReLU is significantly greater than that of SiLU. Besides, the task-specific performance in Table 1 and the trend of training loss in Appendix D reveal the comparable performance between ReLU and SiLU activations. Based on the above observations, **ReLU is more competent as the activation function than SiLU due to three advantages**: an increasing trend of sparsity, significantly higher sparsity ratio, and comparable performance.

### 5.2. Width-Depth Ratio

The width-depth ratio, defined as the ratio of the hidden dimension to the layer number, reflects the shape of a Transformer and is a key architectural property that potentially influences activation sparsity. To inspect its influence on the activation sparsity, we conduct experiments on the 0.1B ReLU-activated model and select 9 different width-depth ratios. The limits of the activation ratio and the training loss are plotted in Figure 5 and Figure 6, respectively.

As demonstrated by Figure 5, **under a bottleneck point (about 114 for 0.1B), the activation ratio linearly increases with the width-depth ratio**. After this bottleneck, the activation ratio fluctuates around 8%. From the sparsity aspect, a smaller width-depth ratio is more helpful. However, Figure 6 demonstrates that an extremely small width-depth ratio causes significant performance degradation. This is attributed to the training instability of deep

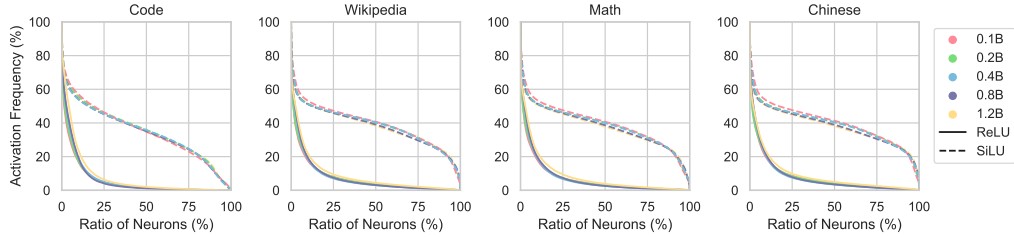

Figure 9: The distribution of the neuron activation frequencies on four datasets within models of distinct scales.

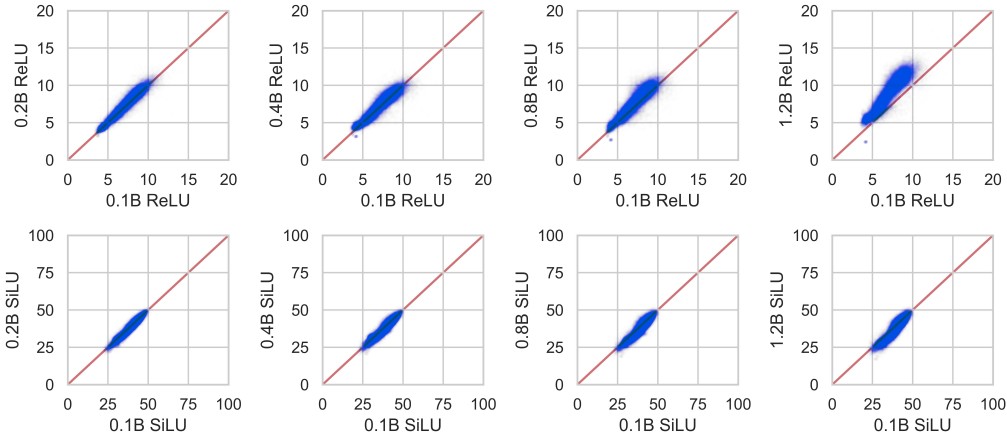

Figure 10: The activation ratio (%) distributions of 71,549 tokens sampled from the vocabulary. We conduct a pair-wise comparison of the average activation ratio of each token within models of different scales. The red line is the $y = x$ curve.

Transformers (Petty et al., 2023; Wu & Tang, 2024). Therefore, the best width-depth ratio should fall on **the smallest point of the interval that ensures stable training and the best performance** (e.g., from 74 to 282 for 0.1B).

### 5.3. Parameter Scale

To obtain comprehensive scaling properties of activation sparsity with the increase of scales (i.e., the number of non-embedding parameters), we obtain the limit of activation ratio on the above pre-trained models with 5 distinct scales but similar width-depth ratios. From the results plotted in Figure 7, we can reach the first observation that **under similar width-depth ratios, the limit of activation ratio, as the amount of pre-training data approaches infinity, is weakly related to the parameter scale**. For SiLU settings, the activation ratio decreases slightly by 2.7 points from 0.1B to 1.2B. By contrast, for ReLU settings, the activation ratio marginally increases by 1.7 points from 0.1B to 1.2B.

To reflect the evolving dynamics of sparsity, we compute the derivatives of the sparsity-data curve as fitted in Section 5.1 and plot the trend of derivatives with the increase of data-scale ratio[2]. The results in Figure 8 clearly demonstrate that

smaller models tend to converge faster than larger models to the limit, as the absolute values of their derivatives are much larger. We try to explain the above observations.

**Observation: Neurons within models of different scales present similar activation patterns.** To support this point, we conduct two experiments from the aspect of *dataset-wise* and *token-wise* activation patterns respectively.

To inspect the dataset-wise activation distribution, we consider four subsets of the pre-training data and compute the distribution of activation frequencies (i.e., the times that a neuron is activated divided by the total number of tokens) among the neurons within models of different scales. As demonstrated by Figure 9, for all datasets, the distribution patterns of neuron activation frequencies are similar across different scales. While this observation holds on average, special cases exist in certain layers (see Appendix J).

For the token-wise activation, we sample 71,549 tokens from the vocabulary and count their activation ratios on a sufficiently large amount of data. Next, we compare the activation ratios of each token among models of different scales in a pair-wise manner. Figure 10 clearly shows that

---

[2]The data-scale ratio means the ratio of the number of training tokens to the parameter scale. We choose this variable as previous

works have demonstrated the roughly proportional relationship between the optimal amount of pre-training data and the parameter scale (Hoffmann et al., 2022; Besiroglu et al., 2024).

most tokens maintain a close activation ratio across models of various scales.

The above two experiments both support the insensitivity of the activation pattern to the parameter scale. This can potentially provide one explanation for why the activation sparsity is quite weakly correlated with the model sizes.

**Assumption: Neurons within models of different scales present similar specialization.** The neurons in FFNs tend to specialize into certain functions during the training progress (Li et al., 2022; Zhang et al., 2023). However, few works have studied how such specialization differs in models of distinct scales. As stated above, both the dataset-wise and token-wise activation patterns are insensitive to the parameter scale. In other words, the numerical distribution of neurons activated for a certain function (e.g., a specific category of datasets or syntactic elements) is similar. Therefore, it is reasonable to assume that the specialization of neurons is also scale-insensitive.

**Deduction: Smaller models converge faster to the limit of activation ratio mainly due to their small amount of neurons.** To simplify this problem, we model the specialization of neurons as a grouping process, where each neuron can be placed into zero or more groups (considering the potential existence of dead neurons and versatile neurons). Suppose the $d_f$ neurons should specialize into $G$ groups, each of them having $t_1, t_2, ..., t_G$ neurons respectively. Based on the assumption of similar activation patterns and neuron specialization, the ratio of neurons placed in each group (i.e., $0 < t_i/d_f \le 1$, $i = 1, 2, ..., G$) should be shared across different parameter scales. We can obtain the number of all the possible grouping results $T(d_f)$ easily,

$$T(d_f) = \prod_{i=1}^{G} C_{d_f}^{t_i} = \prod_{i=1}^{G} \frac{d_f!}{t_i!(d_f - t_i)!}, \quad (6)$$

where $C_{d_f}^{t_i}$ is the combinatorial number, the number of possibilities to select $t_i$ neurons from $d_f$ ones. Obviously, $T(d_f)$ grows in a factorial speed with $d_f$, much faster than the linear function. For larger models, the number of neuron specialization possibilities is significantly greater, and thus more training expenses are required to form stable neuron specialization and approach the limit of activation ratio.

## 6. How can we build a more sparsely activated and efficient LLM?

Based on the above findings, we finally come to the approach to building an LLM with greater activation sparsity: **Use ReLU as the activation function with a larger amount of pre-training data, and a small width-depth ratio within the interval ensuring the training stability.**

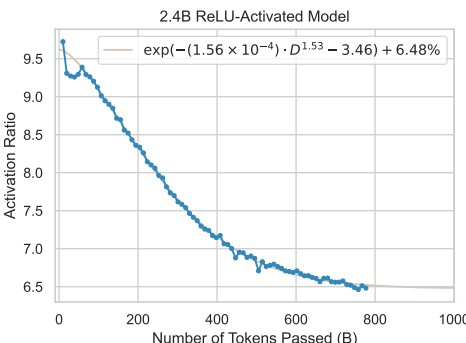

Figure 11: The activation-data trend of the 2.4B ReLU model, with the fitted logspace power-law presented.

To validate the reliability and generalizability of our findings, we train a 2.4B model based on the above experience. Equipped with ReLU activation, near 800B training data, and a width-depth ratio close to the above models (also small enough and close to the smallest point of the training stability interval), this model achieves a low limit activation ratio of 6.48%, quite close to those values of models from 0.1B to 1.2B. Moreover, its activation-data trend can also be well fitted with a decreasing logspace power-law, as shown in Figure 11. These are all consistent with previous findings.

Moreover, to demonstrate the practical acceleration value, we compare the decoding speed of the 2.4B model with PowerInfer (Song et al., 2023) and "llama.cpp" (Gerganov, 2023), respectively. While PowerInfer utilizes sparsity to save computation, "llama.cpp" only conducts dense FFN computation. Consequently, the former achieves $4.1\times$ speedup compared with the latter, revealing the potential of activation sparsity for efficient LLMs. Refer to Appendix L for experiment details.

## 7. Discussion

In this section, we mainly discuss other potential values of this work, especially in terms of **monitoring the model dynamics from new aspects**.

**Training-time predictable sparsity** Our work enables the prediction of sparsity during the training stage. By fitting the power-law or logspace power-law between activation sparsity and the amount of pre-training data, model developers can either predict the theoretical upper/lower bound sparsity of a model to evaluate its potential (e.g., in inference acceleration), or estimate the number of tokens required to achieve a desired sparsity ratio.

**Lens for the convergence of neuron specialization** Generally, the loss curve is an important signal of the training state (e.g., at which point the model converges). However, if we compare the loss curve in Figure 13 and the sparsity

(activation ratio) curve in Figure 4, we will find that **the convergence speed of activation sparsity is much slower than loss, indicating the ongoing of neuron specialization even when the loss changes little**. Despite the wide recognition of the neuron specialization phenomenon (Li et al., 2022; Zhang et al., 2023), it is unclear when such specialization converges and how to inspect this progress. Besides, the loss curve is often not a good inspector for convergence, especially for LLMs with trillion-level pre-training data. We believe that **the trend of activation sparsity can provide a new lens for inspecting the progress of neuron specialization and training convergence**.

## 8. Conclusion

In this work, we address three important research questions about activation sparsity. First, we prove CETT-PPL-1% as a good sparsity metric with the most accurate recognition of weakly-contributed neurons and negligible performance degradation. Next, quantitative studies are conducted on the influential factors of activation sparsity. Finally, towards a sparser LLM, our findings substantiate the advantage of ReLU activation, more training data, and a small width-depth ratio. We also observe and explain the scale insensitivity of sparsity. These can better instruct LLM developers to build sparser LLMs and leverage their merits.

## Impact Statement

This paper presents work whose goal is to advance the field of large language models and address underexplored issues of activation sparsity. There are many potential societal consequences of our work, none of which we feel must be specifically highlighted here. For more rigorous studies, we also discuss the limitations of our work in Appendix A.

## Acknowledgments

This work is supported by the high-quality development project of MIIT, the National Natural Science Foundation of China (No. 62236004), and a grant from the Guoqiang Institute, Tsinghua University.

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

# A. Limitations

One drawback of our study is the absence of computation (e.g., FLOPS) in some analyses, especially the experiments for width-depth ratios. In Section 5.2, we find a smaller width-depth ratio potentially produces a sparser model. However, with the substantial increase in the number of layers, the training efficiency is significantly decreased, as we have observed in the training process. Therefore, in addition to performance, the computation costs of a model also deserve consideration. Similarly, considering the values of activation sparsity in acceleration, it may be interesting to involve the inference computation as a variable in our study.

Another limitation of our CETT-PPL-$p\%$ metric (as well as all the sparsity metrics relying on a validation dataset) is the sensitivity to different data distributions. Intuitively, the same model can have different sparsity ratios on distinct datasets or tasks. The correlation between sparsity and influential factors (e.g., the form of power laws) can also have dataset-unique characteristics. A piece of evidence already presented in our paper is in Table 1, where the performance on commonsense reasoning tasks is insensitive to $p\%$, largely different from the results on reading comprehension tasks. Moreover, the data mixing policies for pre-training can also have a considerable impact on the activation sparsity, which we leave for future work.

# B. Activation Sparsity and Mixture-of-Experts

Mixture-of-experts (MoE) is a mainstream method to achieve high activation sparsity by introducing constraints at the model architecture level. Typically, MoE uses a token-level top-k parameter selection router to assign a fixed sparsity ratio for each token at each layer (Fedus et al., 2022; Zoph et al., 2022). However, these constraints often sacrifice model flexibility and performance. Recent works reveal the potential performance degradation caused by such inflexible sparsity assignment (Huang et al., 2024; Liu et al., 2024b). Moreover, to inspect the impact of such constraints, we plot the PPL-activation (PPL denotes perplexity) Pareto curve of MoE in Figure 12 and compare it with a vanilla decoder-only Transformer (Touvron et al., 2023) of the same parameter scale and amount of pre-training data[3]. MoE has a significantly worse performance-sparsity trade-off. Moreover, the best sparsity ratio is also hard to predefine, since a too-high or too-low sparsity ratio may lead to more severe performance degradation or substantial unnecessary computation, respectively.

Therefore, to avoid negative impacts on flexibility and performance, in this work, we focus on the intrinsic activa-

---

[3]MoE models of different sparsity are obtained by tuning the number of activated experts, while for the vanilla setting, we adjust the CETT hyper-parameter proposed by Zhang et al. (2024a).

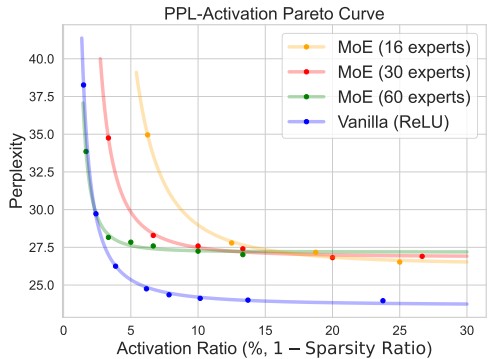

Figure 12: The PPL-activation Pareto curve of the 0.1B MoE with different expert numbers versus the 0.1B vanilla decoder-only Transformer.

tion sparsity within vanilla decoder-only Transformer-based LLMs in this paper, such as GPT (Brown et al., 2020) and LLaMA (Touvron et al., 2023).

# C. Activation Sparsity and Weight Pruning

Weight pruning, also a popular paradigm for inference acceleration, accelerates LLMs by removing certain elements from the model parameters (e.g., neurons, weights, or structured blocks). Nevertheless, the sparsity introduced by weight pruning is fundamentally different from activation sparsity, as the pruning sparsity is completely static, namely independent of inputs. Specifically, weight pruning always drops the same part of parameters regardless of inputs. Consequently, enforcing high such static sparsity can easily cause considerable performance degradation (Frantar & Alistarh, 2023; Xia et al., 2023).

By contrast, activation sparsity dynamically determines weakly-contributed neurons given specific input tokens, which potentially sacrifices less LLM capacity and performance. For example, ReLU-activated models, with considerably higher activation sparsity, have comparable performance to mainstream SiLU-activated ones. Moreover, weight pruning and activation sparsity can also be combined to further promote LLM efficiency.

# D. Training Loss Dynamics

To present the comprehensive training dynamics, we plot the trend of loss with the increase of training data in Figure 13. As can be clearly observed, larger models have smaller training loss. Besides, we also plot the limit values of the training loss with infinite training tokens, shown in Figure 14. As demonstrated in the above two figures, SiLU and ReLU models are well comparable from the loss aspect.

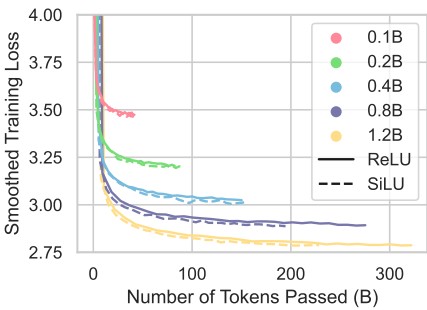

Figure 13: The trend of pre-training loss for models with different scales and activations.

Figure 14: The limits of the training loss with the amount of pre-training data approaches infinity.

## E. Sparsity Stabilizing Strategy

We find that the stochastic factors in gradient descent during pre-training have a significant impact on the metric of activation sparsity. Especially, during the early training stage, the model is far from convergence with considerable sparsity fluctuations, and the magnitude-based sparsity metric can become unstable and cause problems to curve fitting.

To eliminate the influence of these unstable factors to facilitate a smoother sparsity metric, we first drop the sparsity points during the warmup stage for curve fitting. Moreover, we mainly apply the CETT-PPL-$p\%$ on the last several checkpoints (specifically, the last five pre-trained checkpoints) as a whole, binary-searching the hyper-parameter value that controls the average PPL of these checkpoints to just increase by $p\%$. Then this hyper-parameter is applied to all the checkpoints of this pre-training process to measure the sparsity ratio.

## F. Binary Search Algorithm for CETT

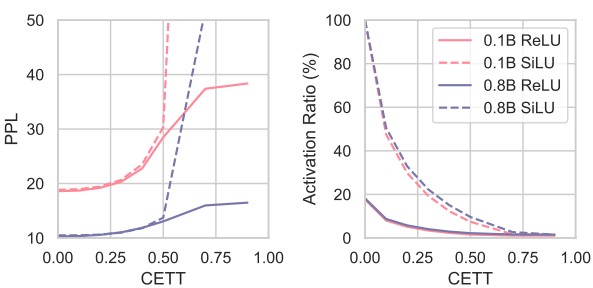

Figure 15: Experiments show that both PPL and the activation ratio change monotonously with CETT, the key hyper-parameter of CETT-PPL-$p\%$. This provides rationality for using a binary search to determine the CETT hyper-parameter.

Given a list of checkpoints, a validation dataset, and a hyper-

parameter $p\%$, we employ Algorithm 1 to find the CETT hyper-parameter that just makes the average PPL of these checkpoints on the validation dataset rise by exactly $p\%$, compared to the dense setting with all the neurons activated. The rationality of using binary search is substantiated by the monotonous relationship between PPL and CETT, as shown in Figure 15. Note that this algorithm can be applied to either a single checkpoint or multiple checkpoints, as adopted in the strategy described in Appendix E.

---

**Algorithm 1** Find the CETT hyper-parameter for CETT-PPL-$p\%$ sparsity

---

**Input:** The input list of checkpoints $CkptList$.
**Input:** The validation dataset $VS$.
**Input:** The PPL increase tolerance $p\%$.
**Input:** The error tolerance $eps$ for binary search.
**Output:** The hyper-parameter $CETT$ that just makes the average PPL of $CkptList$ on $VS$ rise by $p\%$.

---

$l \leftarrow 0,\ r \leftarrow 1$
**while** $r - l > eps$ **do**
   $mid \leftarrow (l + r)/2$
   $PPLRatioList \leftarrow [\,]$
   **for** $Ckpt \in CkptList$ **do**
      $loss_{dense} \leftarrow L_{dense}(Ckpt, VS)$
      $loss_{sparse} \leftarrow L_{sparse}(Ckpt, VS, cett = mid)$
      $PPLRatio \leftarrow \exp(loss_{sparse} - loss_{dense})$
      $PPLRatioList.append(PPLRatio)$
   **end for**
   $MeanPPLRatio \leftarrow \mathrm{Mean}(PPLRatioList)$
   **if** $MeanPPLRatio < 1 + p\%$ **then**
      $l \leftarrow mid$
   **else**
      $r \leftarrow mid$
   **end if**
**end while**
$CETT \leftarrow (l + r)/2$
**return** $CETT$

---

Table 2: Coefficients of activation-data (logspace) power-laws obtained from curve fitting. The curves of ReLU-activated and SiLU-activated models follow Eq. (4) and Eq. (5) respectively.

|      |      | $\alpha$ | $b$ | $c$ | $A_0$ |
|------|------|----------|-----|-----|-------|
| ReLU | 0.1B | $1.01 \times 10^{-01}$ | $-1.51 \times 10^{-02}$ | $3.20 \times 10^{+00}$ | $6.14 \times 10^{-02}$ |
|      | 0.2B | $4.49 \times 10^{-01}$ | $-3.05 \times 10^{+00}$ | $2.86 \times 10^{-01}$ | $6.74 \times 10^{-02}$ |
|      | 0.4B | $6.83 \times 10^{-01}$ | $-3.46 \times 10^{+00}$ | $7.90 \times 10^{-02}$ | $6.90 \times 10^{-02}$ |
|      | 0.8B | $1.01 \times 10^{+00}$ | $-3.49 \times 10^{+00}$ | $7.97 \times 10^{-03}$ | $7.21 \times 10^{-02}$ |
|      | 1.2B | $1.33 \times 10^{+00}$ | $-3.89 \times 10^{+00}$ | $9.03 \times 10^{-04}$ | $7.82 \times 10^{-02}$ |
|      | 2.4B | $1.53 \times 10^{+00}$ | $-3.46 \times 10^{+00}$ | $1.56 \times 10^{-04}$ | $6.48 \times 10^{-02}$ |
| SiLU | 0.1B | $4.79 \times 10^{-01}$ | - | $1.02 \times 10^{-01}$ | $4.09 \times 10^{-01}$ |
|      | 0.2B | $8.44 \times 10^{-01}$ | - | $2.08 \times 10^{-01}$ | $3.90 \times 10^{-01}$ |
|      | 0.4B | $1.03 \times 10^{+00}$ | - | $4.20 \times 10^{-01}$ | $3.85 \times 10^{-01}$ |
|      | 0.8B | $9.95 \times 10^{-01}$ | - | $5.62 \times 10^{-01}$ | $3.83 \times 10^{-01}$ |
|      | 1.2B | $9.67 \times 10^{-01}$ | - | $5.38 \times 10^{-01}$ | $3.82 \times 10^{-01}$ |

Table 3: Hyper-parameters across various parameter scales.

| Parameter Scale | 0.1B | 0.2B | 0.4B | 0.8B | 1.2B | 2.4B |
|-----------------|------|------|------|------|------|------|
| # non-embedding parameters | $1.08 \times 10^8$ | $2.41 \times 10^8$ | $4.52 \times 10^8$ | $7.60 \times 10^8$ | $1.18 \times 10^9$ | $2.44 \times 10^9$ |
| batch size | $3.27 \times 10^5$ | $5.90 \times 10^5$ | $7.86 \times 10^5$ | $1.18 \times 10^6$ | $1.57 \times 10^6$ | $2.10 \times 10^6$ |

# G. Fitting Algorithm and Results

We employ the Levenberg-Marquardt method (Marquardt, 1963) to fit the activation-data curves. To improve the stability of curve fitting, we divide the number of tokens passed (i.e., the amount of pre-training data) by $10^9$ to normalize its magnitude. All the results we obtained from fitting Eq. (4) (for ReLU-activated models) and Eq. (5) (for SiLU-activated models) are shown in Table 2.

# H. Datasets and Benchmarks

**Training data** The pre-training data is a mixture of various corpus, including a cleaned version of Common-Crawl, Dolma (Soldaini et al., 2024), C4 (Raffel et al., 2020), Pile (Gao et al., 2020), the Stack (Kocetkov et al., 2022), StarCoder (Li et al., 2023), and other collected raw corpus. In contrast, the decay data contains additional instruction-tuning data, such as UltraChat (Ding et al., 2023), SlimOrca (Colombo et al., 2024), OssInstruct (Wei et al., 2024), EvolInstruct (Xu et al., 2023), and other collected datasets.

**Validation data** To measure the CETT-PPL-$p\%$ sparsity more precisely, we introduce a tiny validation dataset, which shares the same distribution as the pre-training data. We conduct deduplication to eliminate any intersections between validation and pre-training data.

**Evaluation benchmarks** To evaluate the task-specific performance of models, we introduce the following two groups of benchmarks: (1) *Commonsense reasoning*: We compute the average 0-shot accuracies on PIQA (Bisk et al., 2020), SIQA (Sap et al., 2019), HellaSwag (Zellers et al., 2019), WinoGrande (Sakaguchi et al., 2020), and COPA (Roemmele et al., 2011). (2) *Reading comprehension*: We report the average 0-shot accuracies on BoolQ (Clark et al., 2019), LAMBADA (Paperno et al., 2016), and TyDi QA (Clark et al., 2020).

We also evaluate our model on more complex tasks but fail to obtain performance above the random level. These include: the average pass@1 scores on HumanEval (0-shot) (Chen et al., 2021) and MBPP (3-shot) (Austin et al., 2021), the average accuracies on GSM8K (8-shot) (Cobbe et al., 2021), MMLU (5-shot) (Hendrycks et al., 2020), Big Bench Hard (BBH) (3-shot) (Suzgun et al., 2022), and AGI-Eval (0-shot) (Zhong et al., 2023).

# I. Detailed Training Settings

We utilize the $\mu P$ Transformer (Hu et al., 2024) architecture and adopt its hyper-parameter policies, along with the WSD learning rate scheduling method. Across all parameter scales, the ratio of $d_f$ to $d_h$ is equal to 2.5 consistently, the number of query heads always matches that of key and value heads, and the width-depth ratios range from 48 to 56, generally similar across different scales. The specific number of parameters of various settings are shown in Table 3. We employ the following pre-training hyper-parameters across all

settings: peak learning rate $lr = 0.01$, $\beta_1 = 0.9$, $\beta_2 = 0.95$, $weight\ decay = 0.1$. The batch size depends on the parameter scale, as presented in Table 3.

## J. Dataset-wise Activation Pattern

Although the overall distribution patterns of activation frequencies are similar in terms of the average scenario, they exhibit differences when focusing on neurons in specific layers, such as the first, the last, or the exact middle layer. As shown in Figure 16, models with varying parameter scales have diverse neuron activation frequency distributions in the first layer and the last layer, while the patterns on the middle layer are still largely scale-insensitive.

## K. Performance on Independent Benchmarks

In Table 1, we already provide the average performance on the two groups of commonsense reasoning and reading comprehension. In this section, we present the evaluation scores on independent benchmarks of these two task groups, as shown in Table 4 and Table 5, respectively. From these tables, it can be observed that in commonsense reasoning benchmarks, as the number of model parameters increases from 0.1B to 1.2B, the average evaluation score of the ReLU settings rises from 49.6 to 60.0, while the average score of the SiLU settings increases from 49.5 to 59.6. Similarly, in reading comprehension benchmarks, the score of ReLU settings goes from 28.2 to 53.2, and the score of SiLU settings rises from 27.7 to 54.8. Additionally, models with these two distinct activation functions demonstrate comparable performance at the same parameter scale. Moreover, under the CETT-PPL-1% setting, the models are generally comparable to the dense setting with all neurons activated, whereas under the CETT-PPL-5% setting, they tend to suffer from significant performance on reading comprehension tasks, but the commonsense reasoning scores almost remain unaffected, which is a phenomenon worth studies.

We also evaluate our models on several more complex tasks. However, due to the limited number of parameters, we are unable to obtain reliable results above the random level. The evaluation results for this part are shown in Table 6.

## L. Details of Acceleration Experiments

To demonstrate the practical inference acceleration values of activation sparsity, we run experiments with the 2.4B ReLU-activated model on two different acceleration frameworks: PowerInfer (Song et al., 2023) and "llama.cpp" (Gerganov, 2023). Specifically, PowerInfer, tailored for activation sparsity, involves an offline profiler and online activation predictors to forecast the activation pattern of each neuron. Therefore, PowerInfer can wisely allocate hardware resources according to the activation frequencies of different neurons and save redundant computation and time wasted on weakly-contributed neurons. By contrast, "llama.cpp" does not utilize activation sparsity for acceleration, computing the FFNs in a dense manner.

Both frameworks are compiled with CUDA enabled and run on the same machine with 104 CPUs and 1 NVIDIA A800 GPU. Although "llama.cpp" does not support ReLU and thus cannot correctly conduct inference with our 2.4B model, this does not impact the acceleration experiment as the FLOPS remain the same as a SiLU-activated model. We use 100 test prompts sampled from C4[4], and each prompt is composed of 5 prefix tokens.

Consequently, we found that PowerInfer can perform decoding at an average speed of 41.79 tokens per second, while "llama.cpp" can only reach 10.23 tokens per second. The $4.1\times$ speedup of PowerInfer provides strong evidence of the acceleration potential offered by activation sparsity.

---

[4]https://huggingface.co/datasets/allenai/c4

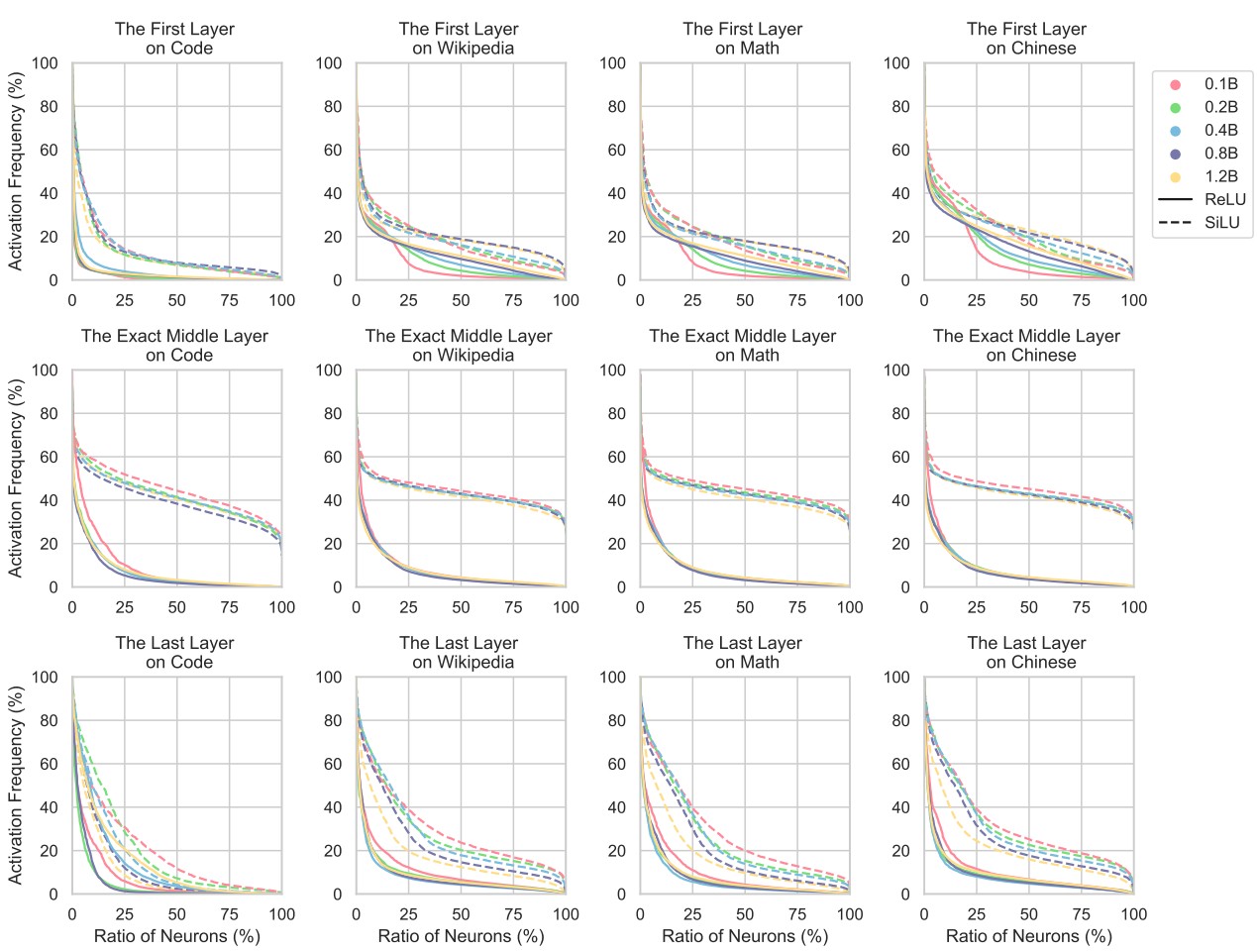

Figure 16: The distributions of average activation frequencies across three individual layers at different positions within models of distinct scales, including four datasets from the pre-training data.

Table 4: Evaluation scores (%) on *commonsense reasoning* benchmarks.

| | | | PIQA | SIQA | HellaSwag | WinoGrande | COPA | Avg. |
|---|---|---|---|---|---|---|---|---|
| | | | acc | acc | acc | acc | acc | |
| 0.1B | ReLU | Dense | 62.8 | 37.8 | 30.5 | 53.0 | 64.0 | 49.6 |
| | | CETT-PPL-1% | 62.7 | 37.4 | 30.5 | 52.6 | 62.0 | 49.1 |
| | | CETT-PPL-5% | 63.1 | 37.6 | 30.3 | 51.1 | 64.0 | 49.2 |
| | | CETT-PPL-10% | 63.0 | 38.0 | 30.5 | 51.5 | 64.0 | 49.4 |
| | SiLU | Dense | 64.3 | 37.6 | 30.9 | 52.8 | 62.0 | 49.5 |
| | | CETT-PPL-1% | 64.3 | 37.5 | 30.7 | 53.0 | 64.0 | 49.9 |
| | | CETT-PPL-5% | 63.5 | 38.4 | 30.5 | 51.5 | 61.0 | 49.0 |
| | | CETT-PPL-10% | 63.8 | 38.1 | 30.4 | 51.3 | 60.0 | 48.7 |
| 0.2B | ReLU | Dense | 66.3 | 38.3 | 37.1 | 53.1 | 65.0 | 52.0 |
| | | CETT-PPL-1% | 66.3 | 38.1 | 37.2 | 52.7 | 64.0 | 51.7 |
| | | CETT-PPL-5% | 66.2 | 38.1 | 37.1 | 52.2 | 65.0 | 51.7 |
| | | CETT-PPL-10% | 66.0 | 37.9 | 37.0 | 51.9 | 65.0 | 51.6 |
| | SiLU | Dense | 67.6 | 39.0 | 37.8 | 51.8 | 65.0 | 52.2 |
| | | CETT-PPL-1% | 68.2 | 39.2 | 37.7 | 52.0 | 65.0 | 52.4 |
| | | CETT-PPL-5% | 67.4 | 38.2 | 37.7 | 51.8 | 65.0 | 52.0 |
| | | CETT-PPL-10% | 66.8 | 38.8 | 37.9 | 52.1 | 64.0 | 51.9 |
| 0.4B | ReLU | Dense | 68.8 | 39.9 | 42.7 | 51.9 | 70.0 | 54.7 |
| | | CETT-PPL-1% | 68.8 | 39.7 | 42.9 | 51.8 | 70.0 | 54.6 |
| | | CETT-PPL-5% | 68.3 | 39.9 | 42.7 | 52.5 | 68.0 | 54.3 |
| | | CETT-PPL-10% | 68.1 | 40.4 | 42.6 | 53.2 | 70.0 | 54.9 |
| | SiLU | Dense | 69.0 | 39.6 | 44.5 | 51.9 | 74.0 | 55.8 |
| | | CETT-PPL-1% | 68.7 | 39.4 | 44.6 | 52.2 | 74.0 | 55.8 |
| | | CETT-PPL-5% | 68.9 | 39.4 | 44.6 | 51.5 | 71.0 | 55.1 |
| | | CETT-PPL-10% | 68.7 | 39.3 | 44.9 | 51.0 | 72.0 | 55.2 |
| 0.8B | ReLU | Dense | 70.1 | 41.8 | 50.4 | 53.6 | 68.0 | 56.8 |
| | | CETT-PPL-1% | 69.8 | 41.8 | 50.2 | 52.8 | 65.0 | 55.9 |
| | | CETT-PPL-5% | 69.9 | 41.8 | 49.7 | 52.3 | 68.0 | 56.3 |
| | | CETT-PPL-10% | 69.6 | 41.8 | 50.0 | 51.8 | 65.0 | 55.6 |
| | SiLU | Dense | 70.4 | 40.9 | 50.6 | 54.0 | 72.0 | 57.6 |
| | | CETT-PPL-1% | 70.3 | 41.4 | 50.6 | 53.9 | 72.0 | 57.6 |
| | | CETT-PPL-5% | 69.9 | 41.3 | 51.0 | 54.1 | 69.0 | 57.1 |
| | | CETT-PPL-10% | 69.5 | 40.7 | 50.6 | 53.2 | 68.0 | 56.4 |
| 1.2B | ReLU | Dense | 71.6 | 44.1 | 57.7 | 56.4 | 70.0 | 60.0 |
| | | CETT-PPL-1% | 71.1 | 44.7 | 58.0 | 55.3 | 69.0 | 59.6 |
| | | CETT-PPL-5% | 70.8 | 43.9 | 57.8 | 54.9 | 69.0 | 59.3 |
| | | CETT-PPL-10% | 70.2 | 43.6 | 57.1 | 53.7 | 72.0 | 59.3 |
| | SiLU | Dense | 71.8 | 41.2 | 57.8 | 56.1 | 71.0 | 59.6 |
| | | CETT-PPL-1% | 71.8 | 40.9 | 57.8 | 57.3 | 70.0 | 59.6 |
| | | CETT-PPL-5% | 71.8 | 41.3 | 57.9 | 55.9 | 67.0 | 58.8 |
| | | CETT-PPL-10% | 71.6 | 41.3 | 58.1 | 55.5 | 70.0 | 59.3 |

Table 5: Evaluation scores (%) on *reading comprehension* benchmarks.

| | | | BoolQ | LAMBADA | TyDiQA | TyDiQA | Avg. |
|---|---|---|---|---|---|---|---|
| | | | acc | acc | F1 | acc | |
| 0.1B | ReLU | Dense | 60.8 | 30.1 | 17.9 | 4.1 | 28.2 |
| | | CETT-PPL-1% | 60.6 | 28.5 | 19.9 | 4.5 | 28.4 |
| | | CETT-PPL-5% | 60.6 | 25.6 | 17.9 | 3.4 | 26.9 |
| | | CETT-PPL-10% | 60.1 | 24.6 | 16.4 | 3.9 | 26.2 |
| | SiLU | Dense | 56.5 | 31.4 | 18.5 | 4.5 | 27.7 |
| | | CETT-PPL-1% | 56.2 | 31.1 | 19.1 | 5.5 | 28.0 |
| | | CETT-PPL-5% | 53.6 | 28.9 | 18.0 | 5.5 | 26.5 |
| | | CETT-PPL-10% | 51.9 | 25.7 | 16.6 | 5.0 | 24.8 |
| 0.2B | ReLU | Dense | 56.3 | 38.4 | 38.0 | 30.0 | 40.7 |
| | | CETT-PPL-1% | 56.2 | 35.8 | 36.8 | 30.0 | 39.7 |
| | | CETT-PPL-5% | 56.4 | 33.0 | 36.3 | 28.6 | 38.6 |
| | | CETT-PPL-10% | 55.9 | 30.8 | 37.4 | 30.2 | 38.6 |
| | SiLU | Dense | 57.5 | 38.7 | 36.3 | 28.2 | 40.2 |
| | | CETT-PPL-1% | 57.5 | 38.3 | 35.3 | 27.5 | 39.6 |
| | | CETT-PPL-5% | 55.2 | 36.0 | 31.6 | 24.3 | 36.8 |
| | | CETT-PPL-10% | 54.5 | 34.0 | 28.1 | 20.9 | 34.4 |
| 0.4B | ReLU | Dense | 61.7 | 42.9 | 43.6 | 28.0 | 44.0 |
| | | CETT-PPL-1% | 61.6 | 41.3 | 42.1 | 26.6 | 42.9 |
| | | CETT-PPL-5% | 60.8 | 39.1 | 39.9 | 23.4 | 40.8 |
| | | CETT-PPL-10% | 60.2 | 37.8 | 39.2 | 22.5 | 39.9 |
| | SiLU | Dense | 57.6 | 43.0 | 41.1 | 25.4 | 41.8 |
| | | CETT-PPL-1% | 56.6 | 43.1 | 40.5 | 23.4 | 40.9 |
| | | CETT-PPL-5% | 55.2 | 39.2 | 38.1 | 20.4 | 38.2 |
| | | CETT-PPL-10% | 52.7 | 35.9 | 35.0 | 17.7 | 35.3 |
| 0.8B | ReLU | Dense | 62.1 | 47.3 | 42.6 | 27.3 | 44.8 |
| | | CETT-PPL-1% | 61.7 | 45.7 | 41.0 | 24.6 | 43.2 |
| | | CETT-PPL-5% | 60.9 | 43.8 | 40.0 | 24.1 | 42.2 |
| | | CETT-PPL-10% | 59.8 | 42.5 | 37.8 | 21.1 | 40.3 |
| | SiLU | Dense | 63.1 | 46.9 | 41.0 | 22.1 | 43.3 |
| | | CETT-PPL-1% | 63.1 | 46.0 | 43.3 | 24.8 | 44.3 |
| | | CETT-PPL-5% | 62.5 | 44.7 | 37.5 | 18.2 | 40.7 |
| | | CETT-PPL-10% | 62.7 | 43.0 | 34.6 | 15.0 | 38.8 |
| 1.2B | ReLU | Dense | 63.3 | 52.5 | 54.3 | 42.5 | 53.2 |
| | | CETT-PPL-1% | 63.4 | 52.2 | 55.0 | 42.7 | 53.3 |
| | | CETT-PPL-5% | 62.1 | 49.5 | 56.3 | 45.2 | 53.3 |
| | | CETT-PPL-10% | 62.6 | 47.7 | 56.8 | 44.5 | 52.9 |
| | SiLU | Dense | 63.2 | 53.4 | 55.2 | 47.3 | 54.8 |
| | | CETT-PPL-1% | 63.7 | 54.2 | 56.1 | 47.5 | 55.4 |
| | | CETT-PPL-5% | 62.2 | 51.2 | 53.1 | 43.9 | 52.6 |
| | | CETT-PPL-10% | 60.2 | 47.5 | 53.1 | 43.4 | 51.1 |

Table 6: Evaluation scores (%) on other more complex benchmarks.

| | | | AGIEval | HumanEval | MBPP | GSM8K | MMLU | BBH | Avg. |
|---|---|---|---|---|---|---|---|---|---|
| | | | acc | pass@1 | pass@1 | acc | acc | acc | |
| 0.1B | ReLU | Dense | 23.4 | 0.6 | 0.3 | 1.8 | 26.3 | 29.3 | 13.6 |
| | | CETT-PPL-1% | 23.3 | 0.6 | 0.3 | 1.7 | 26.5 | 29.5 | 13.7 |
| | | CETT-PPL-5% | 23.5 | 0.6 | 0.1 | 1.9 | 26.3 | 28.7 | 13.5 |
| | | CETT-PPL-10% | 23.4 | 0.0 | 0.2 | 1.4 | 26.4 | 29.7 | 13.5 |
| | SiLU | Dense | 23.6 | 0.6 | 0.8 | 1.6 | 26.1 | 29.2 | 13.7 |
| | | CETT-PPL-1% | 23.5 | 0.6 | 0.4 | 2.1 | 25.6 | 28.5 | 13.4 |
| | | CETT-PPL-5% | 23.6 | 0.6 | 0.3 | 1.4 | 25.8 | 30.6 | 13.7 |
| | | CETT-PPL-10% | 23.0 | 1.2 | 0.4 | 1.4 | 25.8 | 29.0 | 13.5 |
| 0.2B | ReLU | Dense | 23.2 | 2.4 | 1.5 | 1.6 | 27.2 | 28.8 | 14.1 |
| | | CETT-PPL-1% | 22.8 | 2.4 | 1.2 | 2.1 | 26.9 | 30.3 | 14.3 |
| | | CETT-PPL-5% | 22.7 | 2.4 | 1.0 | 1.6 | 27.1 | 29.7 | 14.1 |
| | | CETT-PPL-10% | 23.0 | 2.4 | 1.2 | 2.1 | 26.4 | 30.1 | 14.2 |
| | SiLU | Dense | 24.2 | 4.3 | 1.0 | 2.2 | 25.7 | 29.6 | 14.5 |
| | | CETT-PPL-1% | 24.2 | 4.3 | 1.8 | 2.0 | 25.2 | 29.1 | 14.4 |
| | | CETT-PPL-5% | 23.9 | 5.5 | 1.6 | 1.4 | 25.0 | 29.0 | 14.4 |
| | | CETT-PPL-10% | 23.2 | 3.0 | 0.5 | 2.4 | 24.2 | 28.4 | 13.6 |
| 0.4B | ReLU | Dense | 24.6 | 6.7 | 2.3 | 2.1 | 26.1 | 30.3 | 15.3 |
| | | CETT-PPL-1% | 24.3 | 7.9 | 3.1 | 1.9 | 26.2 | 30.1 | 15.6 |
| | | CETT-PPL-5% | 24.6 | 7.9 | 2.9 | 2.2 | 26.6 | 30.2 | 15.7 |
| | | CETT-PPL-10% | 25.0 | 7.3 | 2.7 | 2.4 | 26.5 | 29.8 | 15.6 |
| | SiLU | Dense | 24.4 | 5.5 | 3.2 | 2.6 | 24.9 | 30.6 | 15.2 |
| | | CETT-PPL-1% | 24.6 | 5.5 | 3.7 | 3.3 | 25.8 | 29.4 | 15.4 |
| | | CETT-PPL-5% | 24.5 | 6.1 | 2.9 | 3.8 | 25.3 | 29.6 | 15.4 |
| | | CETT-PPL-10% | 24.2 | 4.9 | 2.3 | 2.7 | 24.6 | 30.1 | 14.8 |
| 0.8B | ReLU | Dense | 25.4 | 9.2 | 5.3 | 4.2 | 26.3 | 30.1 | 16.7 |
| | | CETT-PPL-1% | 25.7 | 9.2 | 5.8 | 4.5 | 26.3 | 30.0 | 16.9 |
| | | CETT-PPL-5% | 25.3 | 8.5 | 5.4 | 4.5 | 26.5 | 29.8 | 16.7 |
| | | CETT-PPL-10% | 25.8 | 8.5 | 5.0 | 4.0 | 26.4 | 29.2 | 16.5 |
| | SiLU | Dense | 25.4 | 9.2 | 4.7 | 4.1 | 24.7 | 28.9 | 16.1 |
| | | CETT-PPL-1% | 25.1 | 7.9 | 4.6 | 4.0 | 24.8 | 29.7 | 16.0 |
| | | CETT-PPL-5% | 25.1 | 7.3 | 3.8 | 3.6 | 24.5 | 29.4 | 15.6 |
| | | CETT-PPL-10% | 24.8 | 7.3 | 3.9 | 3.0 | 24.2 | 28.8 | 15.3 |
| 1.2B | ReLU | Dense | 26.6 | 7.3 | 6.2 | 6.4 | 33.4 | 29.9 | 18.3 |
| | | CETT-PPL-1% | 26.5 | 9.8 | 7.8 | 7.7 | 33.9 | 30.3 | 19.3 |
| | | CETT-PPL-5% | 25.8 | 7.9 | 7.4 | 6.3 | 34.3 | 30.2 | 18.6 |
| | | CETT-PPL-10% | 25.9 | 7.3 | 6.6 | 5.9 | 34.0 | 30.6 | 18.4 |
| | SiLU | Dense | 26.2 | 9.8 | 9.0 | 5.2 | 32.6 | 30.9 | 18.9 |
| | | CETT-PPL-1% | 27.0 | 11.0 | 8.9 | 5.8 | 32.2 | 30.4 | 19.2 |
| | | CETT-PPL-5% | 25.7 | 7.9 | 8.5 | 5.1 | 31.0 | 30.0 | 18.0 |
| | | CETT-PPL-10% | 25.6 | 9.2 | 6.9 | 4.0 | 30.7 | 30.1 | 17.8 |

