# OpenReview forum: "Sparsing Law: Towards Large Language Models with Greater Activation Sparsity"
_ICML.cc/2025/Conference — ICML 2025 poster_

### Official Review · Reviewer_Po5c · 2025-03-02

**Overall Recommendation:** 4

**Summary:**

This paper tackles activation sparsity in LLMs to boost efficiency. They introduce CETT-PPL-1%, a sparsity metric that keeps perplexity within 1% of dense models, cutting activations. They explore four factors (pre-training data, activation function, width-depth ratio, model scale) across 0.1B to 1.2B models, finding ReLU outperforms SiLU, more data sparsifies ReLU models, deeper models help sparsity up to a limit, and scale barely shifts sparsity caps.

## update after rebuttal
After careful consideration, I have decided to increase the score to "Accept." The lack of formal proof is not an issue at all. Also, I believe the paper has the potential to share valuable insights about increasing the activation sparsity. If the paper includes the references to the missing citations, the paper will give provide even more complete picture.

**Claims And Evidence:**

The claims hold up. CETT-PPL-1% balances sparsity and performance—Figure 2 shows low PPL, and Table 1 keeps downstream scores near dense (avg. drop <0.5%). Factor findings are solid: ReLU beats SiLU (Figure 4), and the 2.4B model’s 93.52% sparsity (Section 6) follows their recipe. Thorough and extensive experiments were done to back the claims.

**Essential References Not Discussed:**

"CATS: Contextually-Aware Thresholding for Sparsity in Large Language Models" and "Training-free activation sparsity in large language models" are simple post-training techniques for SiLU that find a layer-wise absolute threshold to accelerate LLM inference, that were concurrent works to "ReLU^2 wins." Citing this could provide a more complete picture of sparsifying SiLU.

**Experimental Designs Or Analyses:**

Experiments are robust and extensive. A practical limitation would be that the experiment was limited to 2.4B model. Although not feasible, investigating the larger models could have provided more significant insights.

**Methods And Evaluation Criteria:**

CETT-PPL-1% is clever—binary search tunes sparsity per layer works across ReLU and SiLU. They also conduct extensive experiments to find the activation sparsity scaling law. The extensive use of well-established benchmarks is also great.

**Other Comments Or Suggestions:**

N/A

**Other Strengths And Weaknesses:**

While limited to smaller models (<=2.4B models), the paper provides insights into what affects the activation sparsity of language models. It is understandable that it is hard to study a model bigger than that. The experiments presented in the paper already suggests some meaningful insights on what affects activation sparsity.

**Questions For Authors:**

Q1. In Table 1, for a smaller ReLU model, why does CETT1% perform worse than CETT10%?

Q2. Can we extrapolate the scaling law for larger models?

**Relation To Broader Scientific Literature:**

They not only find the best activation sparsity technique, but also go beyond that by investigating the sparsity law. This sets them apart from the prior works that simply present a new sparsification technique. This paper has a potential to serve as a "sparsity scaling law" which can provide more insights into how to train a more sparse model. The paper's in-depth exploration of the four factors that affect activation sparisty is also helpful.

**Theoretical Claims:**

No formal proofs, just fits. ReLU’s logspace power-law (Eq. 4) and SiLU’s power-law (Eq. 5) match Figure 4, with coefficients in Appendix G. Scale-insensitivity leans on similar neuron patterns (Section 5.3), supported by Figures 9-10. Smaller models converging faster (Figure 8) tracks, but the grouping model (Eq. 6) is speculative—no math locks in why sparsity caps across scales.

---

> ### Author Rebuttal · Authors · 2025-03-31
>
> Thank you for your excellent review. These will encourage us to further improve the quality of our work and continuously forge ahead on the research path.
>
> ## Works in "Essential References Not Discussed"
>
> Thank you for pointing out these two works! They are both related to our paper. We discuss these works here, and will cite them in our future versions.
>
> CATS [1] seems to be a post-training sparsification method based on thresholds. However, after carefully reading the paper and codes, we find that CATS is exactly equivalent to the Top-$k$ setting in our experiments in Section 4.1 of our paper. Specifically, for each FFN layer, CATS will find a corresponding cutoff threshold that just controls the sparsity ratio at a desired level. The activations with smaller absolute values will be dropped. Therefore, this is the same as Top-$k$, which reserves the same ratio of activation values for each layer according to the absolute values.
>
> TEAL [2] is a more interesting work. It proposes a completely different paradigm of activation sparsity. Concretely, TEAL focuses on the input sparsity, indicating that for each vector-matrix computation $\mathbf{x}\mathbf{W}^T$, the values at some positions of $\mathbf{x}$ are small, and computation corresponding to these positions can be skipped for acceleration. By contrast, our work focuses on output sparsity, indicating the highly sparse patterns in the output $\sigma(\mathbf{x}\mathbf{W}_{gate}^T)$ of FFN activation functions. Since these two paradigms have completely different definitions of activation sparsity, the sparsity ratios cannot be trivially compared, and the properties (e.g., scaling laws) of them are also different. We are willing to open a new work to study such input sparsity in the future.
>
> ## Lack of Formal Proofs in "Theoretical Claims"
>
> Thank you for pointing out the lack of formal math proofs in our work. We believe that more strict theoretical works are valuable and indispensable in the future. However, at present, LLMs are black-box systems, which are extremely difficult to interpret theoretically. Instead, mainstream works find empirical laws using statistical methods to make the LLM behavior more predictable. Strict math models are hard to build, as the huge parameter scales make LLMs a highly complex system. The study of human brains also encounters similar challenges, and the statistical analysis of signals like brain waves becomes the mainstream paradigm. Still, we admit the value of more rigorous and formal math modeling and will make our analyses more convincing.
>
> ## Question 1
>
> Considering the general ability, the PPL used to measure sparsity is evaluated on a general validation set with the same distribution as the training set, covering a wide variety of corpus. However, a lower average PPL on the general data cannot necessarily ensure better performance on a specific category of downstream tasks. In Table 1, the performance of reading comprehension consistently drops with an increasing PPL, but the scores of commonsense reasoning fluctuate. Such a phenomenon will be more significant for those tasks accounting for a smaller part of the training data.
>
> ## Question 2
>
> Some coefficients in our activation-data power-laws can potentially be extrapolated to larger models.
>
> First, the limit activation ratio $A_0$ is clearly weakly correlated to the parameter scale. This is also an important finding already stated in this article.
>
> Besides, by Table 2, for ReLU-activated models, the coefficient $\alpha$ monotonically increases, while $c$ monotonically decreases with the model scale. Coefficient $b$ seems to have little value fluctuation when the model scale is larger than 0.4B. These indicate that there probably exist quantitative relationships between these coefficients and the model scale, or in other words, we may incorporate the model scale into our scaling law. However, finding a well-fit law including the model scale is too expensive, as dozens of scales of models should be trained for data preparation. Even the most famous work on scaling laws by OpenAI [3] did not cover model scales larger than 2B.
>
>
>
> ## References
>
> [1] Lee, Donghyun, et al. "CATS: Contextually-aware thresholding for sparsity in large language models." *arXiv preprint arXiv:2404.08763* (2024).
>
> [2] Liu, James, et al. "Training-free activation sparsity in large language models." *arXiv preprint arXiv:2408.14690* (2024).
>
> [3] Kaplan, Jared, et al. "Scaling laws for neural language models." *arXiv preprint arXiv:2001.08361* (2020).

---

### Official Review · Reviewer_RPBu · 2025-03-08

**Overall Recommendation:** 4

**Summary:**

This paper investigates activation sparsity in LLMs through extensive experiments. The main findings include:
1) A quantitative analysis of sparsity patterns across model scales and width-depth ratios;
2) The relationship between activation sparsity ratio and data scale;
3) Achievement of a 93.52% sparsity ratio and 4.1× speedup compared to the dense model on a 2.4B parameter model using CETT-PPL-1%.

## update after rebuttal
Thanks for the authors' responses, which address most of my concerns. I have decided to increase the score.

**Claims And Evidence:**

Yes, the sparsity analysis is backed by comprehensive measurements across different model scales.

**Essential References Not Discussed:**

The paper has appropriately cited and discussed the key related works. No significant omissions were found in the literature review.

**Experimental Designs Or Analyses:**

The experimental designs are valid.

**Methods And Evaluation Criteria:**

The evaluation methods are appropriate. However, the evaluation framework for activation sparsity is directly adopted from previous work (CETT), with this study merely finding optimal hyperparameters under a specific experimental setting (PPL increase tolerance of p%). Besides, the reported 4.1× speedup on a 2.4B parameter model lacks convincing evidence for real-world application scenarios.

**Other Comments Or Suggestions:**

While the authors clearly differentiate their work from MoE and parameter pruning, two potential improvements were overlooked: (1) exploring combinations with other optimization techniques could yield more substantial improvements beyond the 4.1× speedup; (2) demonstrating practical applications would better justify the research value.

**Other Strengths And Weaknesses:**

The analysis is comprehensive, with experimental investigations across different model scales and detailed examinations of sparsity patterns. The paper is well-structured.
However, the work primarily extends existing work (CETT) without introducing fundamentally new concepts and focuses more on analysis rather than proposing new solutions.

**Questions For Authors:**

N/A

**Relation To Broader Scientific Literature:**

The work appears to be an extension of CETT's findings on ReLU sparsity. While it provides more comprehensive analysis, it doesn't present fundamentally new insights or directions.

**Theoretical Claims:**

The paper does not present any theoretical proofs for its claims.

---

> ### Author Rebuttal · Authors · 2025-03-31
>
> Thank you for your excellent review. These will encourage us to further improve the quality of our work and continuously forge ahead on the research path.
>
> ## Practical Acceleration using Activation Sparsity
>
> In Section 6, we present an acceleration experiment based on our 2.4B sparsely-activated model, achieving a 4.1$\times$ speedup ratio. This is conducted on the machine with 1 NVIDIA A800 GPU and PowerInfer, a SOTA sparsity-based framework. As pointed out by the reviewer, there may exist gap between this setting and real-world application scenarios. Therefore, we conduct new acceleration experiments under the following setting: (1) We use the device **"NVIDIA Jetson Orin NX", a real-world representative end-side device tailored for AI**. (2) We combine activation sparsity with **Q4 quantization**, a mainstream acceleration technique.
>
> The decoding speeds (token/sec) of the 2.4B model on NVIDIA Jetson Orin NX are shown in the following table:
>
> | Dense (llama.cpp) | Sparse (PowerInfer) | Sparse (PowerInfer+Q4) |
> | :---------------: | :-----------------: | :--------------------: |
> |5.42|8.76|10.97|
>
> Using activation sparsity, we achieve a considerable speedup compared to the dense setting. Moreover, activation sparsity can also be combined with quantization and achieve an even higher decoding speedup.
>
> Finally, we admit that the combination of activation sparsity and some other acceleration methods is non-trivial. For example, we are already working on its combination with speculative decoding. In this case, we use a new auxiliary loss to increase the similarity between activation patterns of neighbor tokens. With such modification, we are able to effectively utilize activation sparsity to further promote the efficacy of speculative decoding.
>
> ## Contributions of This Work
>
> **Our work is not a mere extension of previous work CETT.** Instead, we expect this work to be the foundation of future works involving the **measurement, analysis, and training of sparsely-activated LLMs**. Specifically, we give answers to three important questions:
>
> - How can activation sparsity be measured more accurately?
>
> We present a new perspective on activation sparsity: **sparsity is the function of performance and must be measured under a specific tolerance of performance drop (i.e., PPL increase)**. Most existing works focus on ReLU-based activation sparsity, considering sparsity a fixed value with no connection to performance. After that, "ReLU$^2$ Wins" proposes CETT [1], enabling us to evaluate the sparsity of non-ReLU LLMs, as sparsity is considered a function of CETT.
>
> Admittedly, in terms of methodology, our CETT-PPL-1% mainly introduces a hyper-parameter search process, finding an appropriate CETT value under a PPL increase ratio. This conversion is simple but important, as sparsity is finally linked to performance, and we are able to inspect the sparsity of LLMs under a specific expectation of performance. After all, it is more intuitive and reasonable to measure sparsity from the lens of performance rather than the ambiguous CETT. Otherwise, it will be complicated to compare the sparsity of models with different architectures (e.g., measuring the sparsity at multiple CETT points and comparing the Pareto curves).
>
> - How is activation sparsity affected by the model architecture and training process?
>
> We present a systematic quantitative analysis of the influential factors of activation sparsity, including the activation function, amount of training data, parameter scale, and width-depth ratio. Analytical scaling laws are also found between sparsity and the amount of training data under ReLU and SiLU. These findings are the basis of the third question.
>
> - How can we build a more sparsely-activated and efficient LLM?
>
> As the most important long-term contribution of this work, based on the above findings, **we propose the better approach to obtaining more sparsely-activated LLM**: Use ReLU as the activation function with a larger amount of pre-training data, and a small width-depth ratio within the interval ensuring the training stability. We also pre-train a 2.4B model from scratch, with an extremely low activation ratio of 6.48%, to re-validate our findings. Our work can provide instructional values for designing and pre-training an LLM with greater activation sparsity, which helps produce more efficient LLMs.
>
> ## References
>
> [1] Zhang, Zhengyan, et al. "ReLU$^ 2$ Wins: Discovering Efficient Activation Functions for Sparse LLMs." *arXiv preprint arXiv:2402.03804* (2024).

---

### Official Review · Reviewer_BaNd · 2025-03-13

**Overall Recommendation:** 4

**Summary:**

This paper addresses three main directions related to activation sparsity. First, they introduce a new metric which they show to be better than existing activation sparsity metrics. Then, they explore the relationship between various details of the training process with the ability for a model to achieve high activation sparsity. Finally, they demonstrate that through a combination of various features discovered in the second direction, they are able to train a highly sparse LLM with an activation ratio of 6.48% with minimal performance degradation (ie at most 1% increase in perplexity).

## Update after Rebuttal
No additional comments, the authors answered my questions and I maintain my score.

**Claims And Evidence:**

Yes, for the most part I think the claims are supported by evidence. One additional experiment I would like to see is a side-by-side comparison of LLM training as proposed in section 6 with a baseline and/or ablations of each of the components. I understand this is likely infeasible in the rebuttal period but it would be nice to see a direct comparison of even a much smaller LLM trained from scratch if possible. To be clear, if this is not computationally feasible within the time frame, it will not affect my final score.

**Essential References Not Discussed:**

N/A

**Experimental Designs Or Analyses:**

Yes, the experimental design and analysis seem sound. The paper clearly goes through each component, providing evidence for each claimed relationship to activation sparsity.

**Methods And Evaluation Criteria:**

Yes, methods and evaluation criteria both make sense.

**Other Comments Or Suggestions:**

N/A

**Other Strengths And Weaknesses:**

I think this paper is well structured and written. It has novelty in the proposal of a new metric, it backs up this metric with experimental evidence of its superiority, and provides general principles/guidelines with experimental evidence.

**Questions For Authors:**

N/A

**Relation To Broader Scientific Literature:**

The most relevant tie to previous literature is building upon the CETT metric to develop their proposed metric, CETT-PPL-1%. Otherwise, it seems to contribute a nice set of findings to both the activation sparsity and scaling law literature.

**Theoretical Claims:**

There are no theoretical claims that need to be checked.

---

> ### Author Rebuttal · Authors · 2025-03-31
>
> Thank you for your excellent review. These will encourage us to further improve the quality of our work and continuously forge ahead on the research path.
>
> ## Ablation Studies in "Claims And Evidence"
>
> Thank you for reminding us of the ablation studies on the 2.4B model. As it is really expensive to pre-train a new 2.4B model from scratch, we'd like to re-state the results on smaller models for each ablation factor. Most of these are already presented in the current manuscript.
>
> As mentioned in Section 6, we consider 3 factors: (1) ReLU as the activation function; (2) a larger amount of training data; (3) a small width-depth ratio within the interval ensuring the training stability.
>
> ### Activation Function
>
> For each size among "0.1B, 0.2B, 0.4B, 0.8B, 1.2B", as shown in Figure 7, **replacing ReLU with SiLU can cause a considerable increase in the activation ratio by more than 30%**, indicating worse sparsity. Note that these two activation functions do not have significant performance differences by task performance (Table 1) and training loss (Figure 14).
>
> Specifically, we present the limit activation ratio (%) of the 0.1B/0.2B models with different activation functions in the following table. Clearly, **ReLU has significantly higher sparsity than commonly used SiLU and GELU**.
>
> |      | ReLU | SiLU | GELU |
> | :--: | :--: | :--: | :--: |
> | 0.1B | 6.14 | 40.9 | 33.3 |
> | 0.2B | 6.74 | 39.0 | 34.2 |
>
> ### Amount of Training Data
>
> For each size among "0.1B, 0.2B, 0.4B, 0.8B, 1.2B", as shown in Figure 4, the ReLU-activated models always display lower activation ratios (i.e., higher sparsity ratios) with the increase in the amount of training data. This fact is re-validated in the 2.4B model. Figure 11 indicates that 2.4B has a similar negative relationship between the activation ratio and the amount of training data. Therefore, **a decrease in the amount of training data can cause a sparsity drop for ReLU-activated models**.
>
> ### Width-Depth Ratio
>
> We conduct a systematic study on the relationship between sparsity and the width-depth ratio in Section 5.2. As shown in Figure 5 and Figure 6, for the 0.1B ReLU-activated model, the limit of activation ratios linearly increases with the width-depth ratio before a bottleneck point 114, and the lowest training loss with training stability can be achieved within the interval [74, 282]. The 2.4B model as well as other settings from 0.2B to 1.2B all adopt a width-depth ratio close to the smallest point of the training stability interval. In 0.1B experiments, this setting is demonstrated to have the lowest activation ratio under the premise of training stability. **A smaller width-depth ratio than this can cause training instability and worse performance, and a larger one brings about a sparsity drop.**

---

### Decision · Program_Chairs · 2025-05-01

**Decision:**

Accept (poster)

**Comment:**

All reviewers recommend accepting the paper. The proposed CETT-PPL-1% is quite nice and seems effective. Some concerns were raised about not finding anything fundamentally new on top of CETT, because CETT-PPL-1% is just a hyperparameter search. But I think that moving towards practical and deployable activation sparsity is quite important, and this work does just that.

I would recommend to the authors to revise and update the related work section to include a discussion of where this work fits in the broader catalogue of sparsity. Someone who reads this paper will want to know whether they can combine TEAL and CETT-PPL-1%, and you are the best people to answer this question proactively.